# Rapid quantification assay of hepatitis B virus DNA in human serum and plasma by Fully Automated Genetic Analyzer µTASWako g1

**Moto Watanabe[1], Hidenori Toyoda[2], Tomohisa Kawabata[3]***

**1** Diagnostics Research Laboratories, FUJIFILM Wako Pure Chemical Corporation, Amagasaki, Hyogo, Japan, **2** Department of Gastroenterology and Hepatology, Ogaki Municipal Hospital, Ogaki, Gifu, Japan, **3** Department of Diagnostics Development, FUJIFILM Wako Pure Chemical Corporation, Chuo-Ku, Tokyo, Japan

* tomohisa.kawabata@fujifilm.com

## Abstract

Real-time monitoring of serum hepatitis B virus (HBV) levels is essential for the management of patients with chronic HBV infection in clinical practice, including monitoring the resistance of anti-HBV nucleotide analog or the detection of HBV reactivation. In this context, serum HBV deoxyribonucleic acid (DNA) quantification should be rapidly measured. A rapid HBV DNA quantification assay was established on the Fully Automated Genetic Analyzer, µTAS-Wako g1. The assay performs automated sample preparation and DNA extraction, followed by the amplification and detection of quantitative polymerase chain reaction (PCR) combined with capillary electrophoresis (qPCR-CE) on integrated microfluidic chip. This study aimed to evaluate the analytical and clinical performance of HBV DNA assay on the µTASWako g1 platform in human serum and EDTA-plasma. The HBV DNA assay has a linear quantitative range from 20 to $10^8$ IU/mL of HBV DNA with standard deviation (SD) of $\leq 0.14$ $\log_{10}$ IU/mL. The limits of detection of the assay were 4.18 for the serum and 4.35 for EDTA-plasma. The HBV assay demonstrated the equivalent performance in both human serum and EDTA-plasma matrices. The HBV genotypes A to H were detected with an accuracy of ±0.34 $\log_{10}$ IU/mL. In quantification range, the HBV DNA assay was correlated with Roche cobas Ampli-Prep/cobas TaqMan Ver2.0 (CAP/CTM v2) ($r = 0.964$). The mean difference (µTASWako g1–CAP/CTM v2) of the reported HBV DNA was −0.01 $\log_{10}$ IU/mL. Overall, the sensitivity, accuracy, and precision of the µTASWako g1 HBV assay were comparable to the existing commercial HBV DNA assay, and the assay can be completed within 110 min. This evaluation suggests that the HBV DNA assay on the µTASWako g1 is potentially applied for alternative method of the HBV viral load test, in particular with the advantage of the HBV DNA result availability within 2 h, improving the HBV infection management.

**Data Availability Statement:** All relevant data are within the paper and its Supporting Information file.

**Funding:** Specific grant numbers; None Initials of authors who received each award; M.W Full names of commercial companies that funded the study or authors; Fujifilm Wako Pure Chemical Corporation Initials of authors who received salary or other funding from commercial companies; M.W and T.K URLs to sponsors' websites; https://www.fujifilm. com/ffwk/en The funder provided support in the form of salaries for authors M.W and T.K, but did not have any additional role in the study design, data collection and analysis, decision to publish, or preparation of the manuscript. The specific roles of these authors are articulated in the 'author contributions' section.

**Competing interests:** The authors have declared that no competing interests exist.

## Introduction

Hepatitis B virus (HBV) specifically infects the hepatocytes and causes severe liver diseases [1, 2]. Chronic HBV infection was associated with significant risks for the occurrence of cirrhosis and hepatocellular carcinoma, the major causes of death associated with HBV infection [3–5]. The World Health Organization (WHO) reported that approximately 257 million people were chronically infected with HBV in 2015 [3]. Serum HBV DNA measurement is a pivotal test for the management of patients with HBV, including the assessment on the risk for liver flare and, in particular, monitoring the efficacy of antiviral therapy [6].

In addition to chronic infection, HBV is potentially reactivated in some patients. HBV reactivation occurs in patients who undergo chemotherapy or immunosuppressive therapies and often causes life-threatening liver failure. Reactivation occurs not only in HBsAg-seropositive patients but also in those with resolved HBV infection who are seronegative for HBsAg but seropositive for antibody against hepatitis B core antigen (anti-HBc) and/or antibody against HBsAg (anti-HBs) [7–11]. To prevent HBV reactivation-related hepatitis, HBV DNA monitoring-guided preemptive antiviral therapy using anti-HBV nucleos(t)ide analogs (NAs) is recommended by several guidelines for patients with resolved HBV infection [12–14]. However, the duration required for HBV DNA measurements, including both qualitative and quantitative, hinders agile and effective preventive strategies in real clinical practice, and HBV DNA measurement technology with rapidly available results.

The HBV DNA measurement in human serum and plasma utilizes quantitative real-time polymerase chain reaction (qPCR)- or TMA-based assay incorporated in commercial systems, such as cobas AmpliPrep/cobas TaqMan HBV Test ver.2.0 (CAP/CTM v2) (Roche Diagnostics, Indianapolis, IN, USA), RealTime HBV assay (Abbott Laboratories, Des Plaines, IL, USA), and Aptima HBV Quant assay (Hologic Inc., San Diego, CA, USA) [15, 16]. These systems are generally designed to be high-throughput tests for batch testing of multiple specimens within a run. Therefore, the turnround time varies and may require until a few days from the time of blood collection to obtain results.

In contrast, HBV RNA testing using µTASWako g1 analyzer can analyze a single sample or up to four samples in parallel with a single run, facilitating a rapid turnover. The analyzer conducts automated sample preparation and nucleic acid extraction, followed by amplification and detection of target amplicons by qPCR combined with capillary electrophoresis (CE) on integrated microfluidic chip [17, 18]. The total assay time on µTASWako g1 is within 45 min for *Mycobacterium tuberculosis* DNA qualitative assay, within 70 min for severe acute respiratory syndrome coronavirus 2 (SARS-CoV-2) RNA qualitative assay. In this study, we established an HBV DNA quantification assay by taking advantage of the capability and usability of the µTASWako g1 platform. This assay can complete the HBV DNA quantification within 110 min. We evaluated the analytical and clinical performance of HBV DNA assay on the µTASWako g1 in human serum and EDTA-plasma to assess whether the assay can be applied as a rapid alternative method for HBV DNA measurement.

## Materials and methods

### HBV primers

Specific amplification of 227-bp HBV DNA is achieved using a pair of primers targeting a conserved region in S gene of HBV among all genotypes (forward primer: 5'-TAMRA-AGACT CGTGGTGGACTTC; reverse primer: GACAAACGGGCAACATAC). The 5'-end of the forward primer was labeled using a fluorescent dye, carboxytetramethylrhodamine (TAMRA), which allowed the HBV amplicon to be labeled for fluorescence detection through electrophoresis.

### WHO international standard and HBV-positive control

The 4th WHO international standard of HBV DNA for NAT (10/266) was obtained from the National Institute for Biological Standards and Control. The HBV-positive control was 3288-bp of the linear DNA fragment amplified by PCR. The construction of HBV-positive control consisted of a sequence from pCR2.1-TOPO and 227-bp of the HBV target sequence derived from HBV serotype *adw* (#45020D, American Type Culture Collection, VA, USA). The HBV target sequence is amplifiable by the set of HBV primer for the assay. The HBV-positive control concentration (unit: IU/mL) was determined based on the 4th International WHO Standard for HBV DNA.

### Internal control (IC) for the HBV assay

IC is 3426-bp of the linear DNA fragment amplified by PCR, which contains 365-bp of the IC target sequence. Both ends of the IC target sequence are composed of HBV target sequence for primers (5' end of the IC target is HBV forward primer sequence and 3' end of the IC target is HBV reverse primer sequence), which allows co-amplification of both HBV and IC target using one set of the HBV primer in the PCR. IC (approximately 1200 copies/assay) was added to the sample at the beginning of the sample preparation to monitor the assay process, including DNA extraction from specimens and PCR amplification and detection. The amount of IC added into the assay was optimized to reproducibly detect the entire quantitative range of the assay without interfering HBV target amplification. The IC concentration (unit: copies/mL) was determined based on the concentration measured by absorbance at 260 nm.

### HBV genotype samples

The 1st international reference panel for HBV genotypes for nucleic acid amplification technique-based assays (5086/08) was obtained from Paul-Ehrlich-Institut. The HBV genotype H-positive plasma specimen (Product#203589) was obtained from SeraCare Life Sciences, Inc (Milford, MA, USA).

### HBV-negative serum and EDTA-plasma

A total of 160 individual HBV-negative serum and 60 individual HBV-negative EDTA-plasma specimens were obtained from BIOIVT (Hicksville, NY, USA). Pooled HBV-negative serum and EDTA-plasma specimens were obtained from BIOIVT (Hicksville, NY, USA).

### Clinical samples from HBV-infected individuals

The assay was examined using 207 clinical serum samples from patients with chronic HBV infection obtained at Ogaki Municipal Hospital, Japan. HBV infection had been confirmed by HBV antigens and HBV DNA testing for all patients. The HBV DNA levels of 207 samples had been measured using cobas AmpliPrep/cobas TaqMan HBV Test ver 2.0, ranging from negative (target not detected), <20 IU/mL (positive and not quantifiable) to $1.7 \times 10^8$ IU/mL. Among the 207 samples, 50 were obtained from patients during treatment with nucleoside analogs for anti-HBV therapy. The HBV DNA negative, which comprised a total of 16 samples, were obtained from patients undergoing nucleoside analog treatment. Samples were collected from 97 males and 110 females, with a median age of 58 years. Patient background is shown in S1 Table. The study protocol on clinical serum samples complied with the Helsinki Declaration and was approved by the institutional review board of Ogaki Municipal Hospital. Written informed consent was waived due to the retrospective use of the stored serum samples. Clinical samples were stored at −80°C and kept frozen during transport with dry ice before testing.

## Automated DNA extraction and sample preparation for qPCR

IC was added to 1 mL of the serum or EDTA-plasma sample. For nucleic acids, the sample was treated with proteinase K and guanidine hydrochloride through incubation at an elevated temperature for 5 min. The released nucleic acids were captured using grass fiber membrane filter assembled in a column with ethanol. Subsequently, the column was washed with a wash buffer containing guanidine hydrochloride, ethanol, and then washed with a wash buffer containing ethanol. The captured DNA was eluted with Tris buffer solution. The eluted DNA solution was mixed with the KOD buffer, dNTPs, KOD exo (-) DNA polymerase (TOYOBO, Osaka, Japan), and HBV primers. A 25 uL of reaction mix was added to the PCR chamber of the microfluidic chip and took approximately 54 min to finish the process from DNA extraction to sample preparation for quantitative polymerase chain reaction (qPCR).

## On-chip quantitative PCR using real-time detection by electrophoresis

A previous study demonstrated experiments of quantitative PCR combined capillary electrophoresis (CE) for real-time detection of target amplicons. Details of instrument setup, the design of an integrated microfluidic chip for qPCR-CE, and the detection methods were described in a previous report [18]. During PCR, air pressure (138 kPa) was used to the PCR chamber and channels through a manifold covering the chamber and reagent wells to suppress evaporation of the reaction mix. Initial denaturation was performed for 30 s at 97˚C. Seven short PCR cycles (3.7 s at 99˚C, 10.3 s at 64˚C, and 14 s at 73.5˚C) were then performed. Afterward, 33 PCR cycles with a longer extension (70 s at 73.5˚C) was performed with one CE injection from the PCR chamber for each subsequent cycle to analyze the accumulated target amplicons and took 56 min to complete the entire qPCR-CE step.

## Data acquisition and analysis

An amplification curve for each amplicon was generated by plotting the peak height (relative fluorescence unit [RFU]) versus the number of PCR cycles. The quantitative cycle (Cq) values were determined based on the intersection between the amplification curve and threshold line on the plots. Cq values were converted to $\log_{10}$ IU/mL based on predetermined calibration curve traceable to the 4th WHO International Standard for HBV DNA. The analysis was conducted using a proprietary software developed using the μTASWako g1 analyzer.

## Statistical analysis

Probit analysis for Limit of detection (LOD) test was performed using Minitab software. Person's correlation coefficients and Passing–Bablok regression analysis were performed using Analyze-it software to confirm the correlation in clinical samples. Bland–Altman analysis was performed to compare methods between μTASWako g1 and CAP/CTM v2 for data obtained from clinical samples.

# Results

## Real-time detection of HBV target and IC amplification by CE

To detect HBV amplification signal by CE using the μTASWako g1 analyzer, the 4th WHO international standard for HBV DNA samples was tested. Fig 1A shows overlay of electropherograms at the PCR cycles from 8th to 40th. The graph exhibits the grown amplicon signals of the HBV target (227-bp) and IC (365-bp). The peak alignment using 300-bp and 500-bp of DNA markers was performed to detect HBV and IC peaks at a specific migration time in electropherograms. A growth curve of HBV was generated by plotting the peak height of the

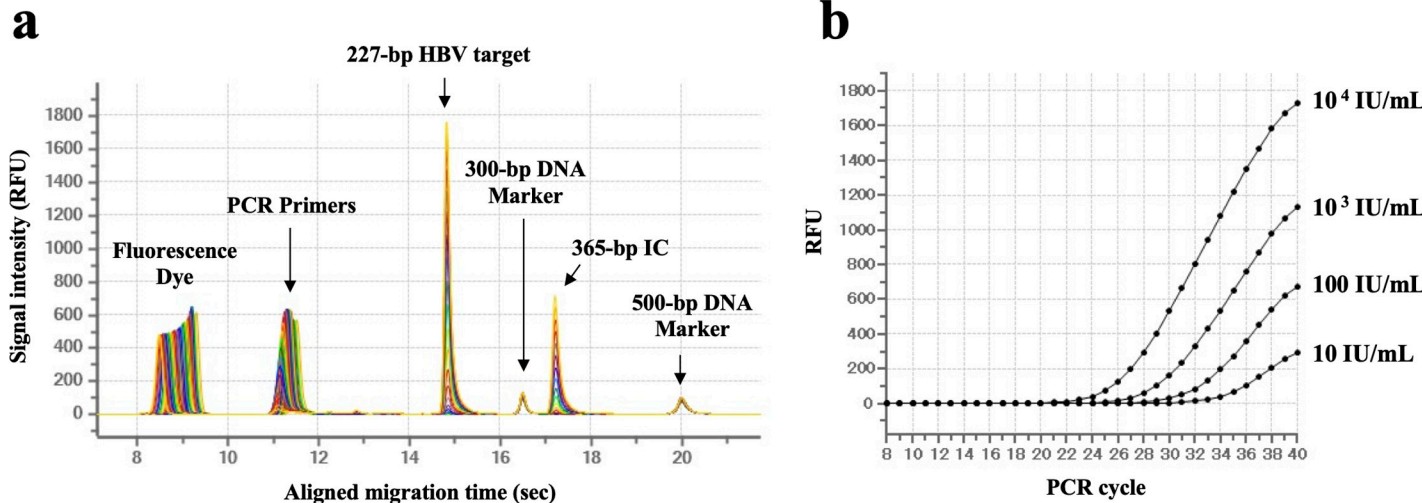

**Fig 1. Real-time detection of HBV target using the µTASWako g1 analyzer.** (a) Overlay of electropherograms in the PCR-CE for $10^4$ IU/mL of the 4th WHO International Standard for HBV. Merged images of 33 electropherograms from 8th to 40th PCR cycle. (b) HBV amplification growth curves. RFU stands for relative fluorescence unit.

amplicon against the number of PCR cycles. Fig 1B shows the composite growth curve tested with 10 to $10^4$ IU/mL of the WHO international standard for HBV.

To assess the linear dynamic range of the HBV assay using qPCR-CE, nine serial dilutions of the HBV-positive control, which is traceable to the 4th WHO International standard, were tested using the µTAS g1 analyzer. In Fig 2A, the HBV Cq plot demonstrated the linear response of HBV Cq values across the range from 10 to $3.5 \times 10^8$ IU/mL of the HBV DNA concentration. The PCR efficiency of HBV target amplification was 98.8%. IC Cq plot is shown in Fig 2B when 10 to $3.5 \times 10^8$ IU/mL of the HBV-positive controls tested. The IC Cq was stably detected within a range (24.5 ± 1), regardless of the variation of HBV concentration in samples. Data indicated that IC was detected as an analyte for monitoring the assay process.

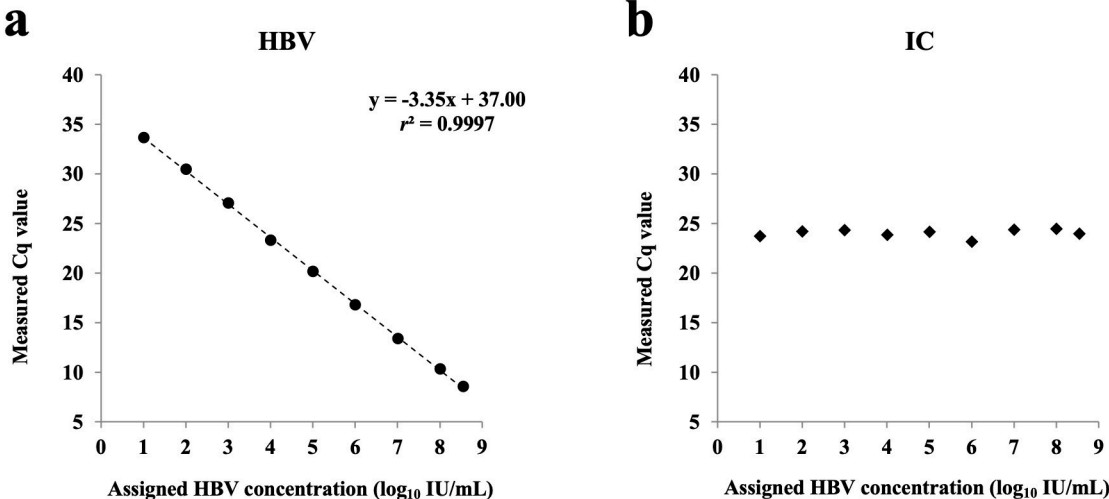

**Fig 2. HBV Cq and IC Cq values plot.** $3.5 \times 10^8$, $10^8$, $10^7$, $10^6$, $10^5$, $10^4$, $10^3$, $10^2$, and 10 IU/mL of the HBV-positive control in the serum were tested using the µTASWako g1 Analyzer. One replicate for each concentration. PCR efficiency of HBV target amplification was 98.8%.

## Analytical performance evaluation

**Sensitivity and specificity.** LOD, defined as the HBV concentration detected with a probability of 95%, was assessed by seven dilutions using the 4th WHO International standard. Dilutions (1.25, 2.5, 5, 7.5, 10, 20, and 50 IU/mL) were prepared in HBV-negative human serum and EDTA-plasma samples, and each dilution was tested in 63 replicates. Probit analysis was performed to determine LOD and 95% confidence interval (CI). Results in Table 1 demonstrate that the HBV assay effectively detected HBV DNA at a concentration of 4.18 IU/mL (95% CI, 3.34–5.92) for 1 mL of the serum and 4.35 IU/mL (95% CI, 3.52–5.99) for 1 mL of EDTA-plasma samples. Results indicated no significant difference in the analytical sensitivity of the HBV assay between serum and EDTA-plasma matrices.

To verify the specificity of the HBV assay, 160 individual HBV-negative serum and 60 individual HBV-negative EDTA-plasma samples were analyzed using the μTASWako g1 HBV assay. No HBV target peak was observed in all samples, and data analysis revealed nonspecific signals at baseline in electropherograms obtained from all samples, which were significantly lower than the detection threshold (30 RFU) in the assay.

**Traceability to the WHO standard and matrix equivalency.** HBV-positive control and 4th WHO international standard were prepared using the HBV-negative serum and EDTA-plasma samples. The concentration range tested for positive control was from $10^2$ to $10^8$ IU/mL, and the WHO standard was from 20 to $10^4$ IU/mL. Each dilution sample was tested in three replicates using the μTASWako g1 analyzer. In Fig 3A, the positive control and WHO standard demonstrated co-linear dilution performance across the linear dynamic range. The measured HBV titers of positive controls and the WHO standard were similar to the expected value (or assigned value) with deviation of ≤0.09 $\log_{10}$ IU/mL. The mean difference of the measured HBV titer between the serum and EDTA-plasma samples was ≤0.05 log IU/mL for positive control and the WHO standard. Fig 3B demonstrates the performance equivalency of the HBV assay between serum and EDTA-plasma matrices.

**Precision (Inter-assay precision).** To assess the assay precision, three dilutions of the positive control and two dilutions of the WHO standard were tested in 21 replicates using the μTASWako g1. Results are shown in Table 2. In the concentration range tested for positive control from $10^2$ to $10^7$ IU/mL, SD of the measured HBV titers were <0.10 $\log_{10}$ IU/mL. In 10 or 20 IU/mL of the WHO standard tested, SD of the measured HBV titers was <0.20 $\log_{10}$ IU/mL. Deviation of the mean titer from the assigned titer was larger (−0.09 $\log_{10}$ IU/mL) when 10 IU/mL of the WHO standard tested.

**Table 1. Limit of detection in the serum and EDTA-plasma using the 4th WHO international standard for HBV.**

| WHO std. concentration IU/mL | No. of replicates | Serum | | | EDTA-Plasma | | |
|---|---|---|---|---|---|---|---|
| | | No. HBV Positive | Detection rate % | LOD* (95% CI) IU/mL | No. HBV Positive | Detection rate % | LOD* (95% CI) IU/mL |
| 50.00 | 63 | 63 | 100 | 4.18 (3.34–5.92) | 63 | 100 | 4.35 (3.52–5.99) |
| 20.00 | 63 | 63 | 100 | | 63 | 100 | |
| 10.00 | 63 | 63 | 100 | | 63 | 100 | |
| 7.50 | 63 | 63 | 100 | | 62 | 98 | |
| 5.00 | 63 | 61 | 97 | | 61 | 97 | |
| 2.50 | 63 | 50 | 79 | | 52 | 83 | |
| 1.25 | 63 | 39 | 62 | | 33 | 52 | |
| 0 | 63 | 0 | 0 | | 0 | 0 | |

* LOD, defined as the HBV concentration detected with a probability of 95%, was determined using Probit analysis.

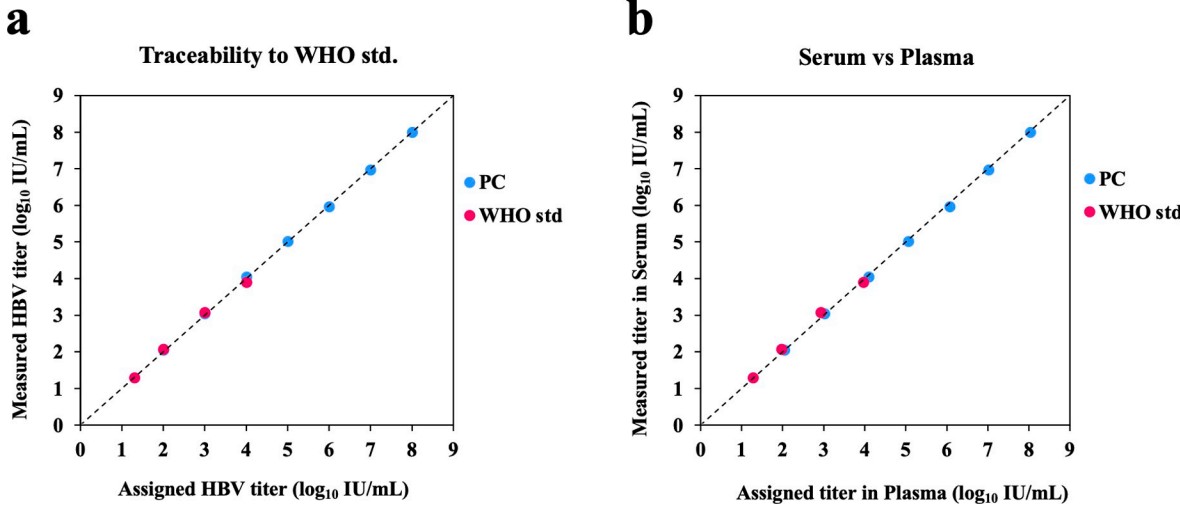

**Fig 3. Traceability to the WHO international standard and matrix equivalency between serum and plasma on μTASWako g1 analyzer.** Dilutions of the positive control (PC) and WHO international standard (WHO std) were tested in three replicates for each concentration. The dots represent the mean $\log_{10}$ transformed titer. The dashed line represents the equality line.

**Genotype accuracy.** To evaluate whether the assay covers HBV genotypes (A, B, C, D, E, F, G, and H), the lower concentration of the 1st WHO International Reference Panel and HBV-positive patient plasma genotype H was tested using the μTASWako g1. Table 3 shows the accuracy of the HBV assay for all genotypes A to H. Measured HBV titers were detected within the accuracy of ±0.34 $\log_{10}$ IU/mL for all genotypes. The absolute mean difference between the measured and assigned titers was ≤0.12 $\log_{10}$ IU/mL. Based on these results, the assay can quantify the HBV DNA with an accuracy of ±0.50 $\log_{10}$ IU/mL for HBV genotypes A to H.

**Correlation with CAP/CTM v2 in clinical samples.** The performance of the μTASWako g1 HBV assay was assessed against the widely used assay, Roche CAP/CTM v2, by testing 207 clinical samples (serum) obtained from HBV-infected patients. Qualitative results from the two assays are categorized in Table 4. The concordance rate of qualitative results was 90.3%. To assess the relationship between HBV DNA levels between the two methods, the Passing–Bablock regression analysis was performed in 157 clinical samples, which reached over the linear dynamic range (20 to $10^8$ IU/mL, 1.3 to 8.0 $\log_{10}$ IU/mL). Fig 4A demonstrates that the measured HBV $\log_{10}$ titers between the two methods are highly correlated across the entire dynamic range ($y = 1.03 \times -0.07$). The correlation coefficiency ($r$) was 0.964 (95% CI: 0.959–0.968). To investigate the measurement bias between the two

**Table 2. Precision of μTASWako g1 analyzer.**

| Assigned HBV DNA concentration | | Source Material | No. of replicates | Precision of μTASWako g1 analyzer | | |
|---|---|---|---|---|---|---|
| IU/mL | $\log_{10}$ IU/mL | | | Mean | SD | CV (%) |
| | | | | ($\log_{10}$ IU/mL) | ($\log_{10}$ IU/mL) | |
| 10,000,000 | 7.00 | Positive control | 21 | 6.98 | 0.05 | 0.73 |
| 10,000 | 4.00 | Positive control | 21 | 4.00 | 0.06 | 1.50 |
| 100 | 2.00 | Positive control | 21 | 1.98 | 0.09 | 4.34 |
| 20 | 1.30 | WHO standard | 21 | 1.28 | 0.14 | 11.00 |
| 10 | 1.00 | WHO standard | 21 | 0.91 | 0.19 | 20.92 |

Note: CV stands for coefficient of variation.

**Table 3. Accuracy for HBV genotypes on µTASWako g1 analyzer.**

| Sample | Genotype | Assigned HBV titer | No. of replicates | Measured HBV titer by µTASWako g1 | |
|---|---|---|---|---|---|
| | | | | Mean (±SD) | Difference* |
| | | $log_{10}$ IU/mL | | $log_{10}$ IU/mL | $log_{10}$ IU/mL |
| WHO Panel #1 | A | 1.70 | 3 | 1.75 (±0.15) | 0.05 |
| WHO Panel #2 | A | 1.70 | 3 | 1.82 (±0.24) | 0.12 |
| WHO Panel #3 | A | 1.70 | 3 | 1.58 (±0.09) | -0.12 |
| WHO Panel #4 | B | 1.70 | 3 | 1.68 (±0.01) | -0.01 |
| WHO Panel #5 | B | 1.66 | 3 | 1.78 (±0.12) | 0.12 |
| WHO Panel #6 | B | 1.70 | 3 | 1.81 (±0.06) | 0.11 |
| WHO Panel #7 | C | 1.70 | 3 | 1.82 (±0.10) | 0.12 |
| WHO Panel #8 | C | 1.70 | 3 | 1.68 (±0.02) | -0.02 |
| WHO Panel #9 | C | 1.70 | 3 | 1.67 (±0.03) | -0.02 |
| WHO Panel #10 | D | 1.70 | 3 | 1.76 (±0.12) | 0.06 |
| WHO Panel #11 | D | 1.70 | 3 | 1.64 (±0.02) | -0.06 |
| WHO Panel #12 | D | 1.70 | 3 | 1.69 (±0.03) | -0.01 |
| WHO Panel #13 | E | 1.70 | 3 | 1.60 (±0.14) | -0.10 |
| WHO Panel #14 | F | 1.70 | 3 | 1.68 (±0.14) | -0.02 |
| WHO Panel #15 | G | 1.70 | 3 | 1.79 (±0.02) | 0.09 |
| SeraCare #203589 | H | 1.60 | 3 | 1.50 (±0.11) | -0.10 |

Note: 1st WHO International Reference Panel for HBV genotypes (WHO Panel) from #1 to #15 were corresponding to genotype A, B, C, D, E, F, and G. Genotype H sample was obtained from SeraCare.

* Difference of HBV titer in log10 IU/mL between the mean measured titer and the assigned titer.

methods, Bland–Altman analysis was performed. The difference in the measured $log_{10}$ titers between CAP/CTM v2 and µTASWako g1 is shown in Fig 4B. The mean difference in $log_{10}$ tier was −0.01 (95% CI: −0.085 to 0.053). The upper and lower limits of agreement were 0.82 and −0.85, respectively. In nine (5.7%) samples, the difference between the two methods were >1.96 times the SD. In 151 (96.2%) samples, the difference between the two methods was ≤1.0 $log_{10}$ IU/mL.

In 30 out of 50 samples, which were either undetectable for HBV DNA or contained <20 IU/mL (<1.3 $log_{10}$ IU/mL) of HBV DNA levels by CAP/CTM v2 and/or µTASWako g1, these qualitative results were concordant between the two methods. Results of the remaining 20 samples are shown in S2 Table. Of the 16 samples, two samples reported as "target not detected" by CAP/CTM v2 were positive for HBV DNA in the µTASWako g1. Conversely, 4 out of 18 samples reported as "<1.3 $log_{10}$ IU/mL" by CAP/CTM v2 were negative for HBV DNA in the µTASWako g1.

**Table 4. Qualitative results of µTASWako g1 HBV assay and CAP/CTM v2 assay.**

| | CAP/CTM v2 assay | | | |
|---|---|---|---|---|
| | Virus detected, | Virus detected, | Virus Not | No. of total |
| | ≥20 IU/mL | <20 IU/mL | Detected | |
| µTASWako g1 HBV assay | | | | |
| Virus detected, ≥20 IU/mL | 157 | 3 | 0 | 160 |
| Virus detected, <20 IU/mL | 11 | 16 | 2 | 29 |
| Virus not detected | 0 | 4 | 14 | 18 |
| No. of total | 168 | 23 | 16 | 207 |

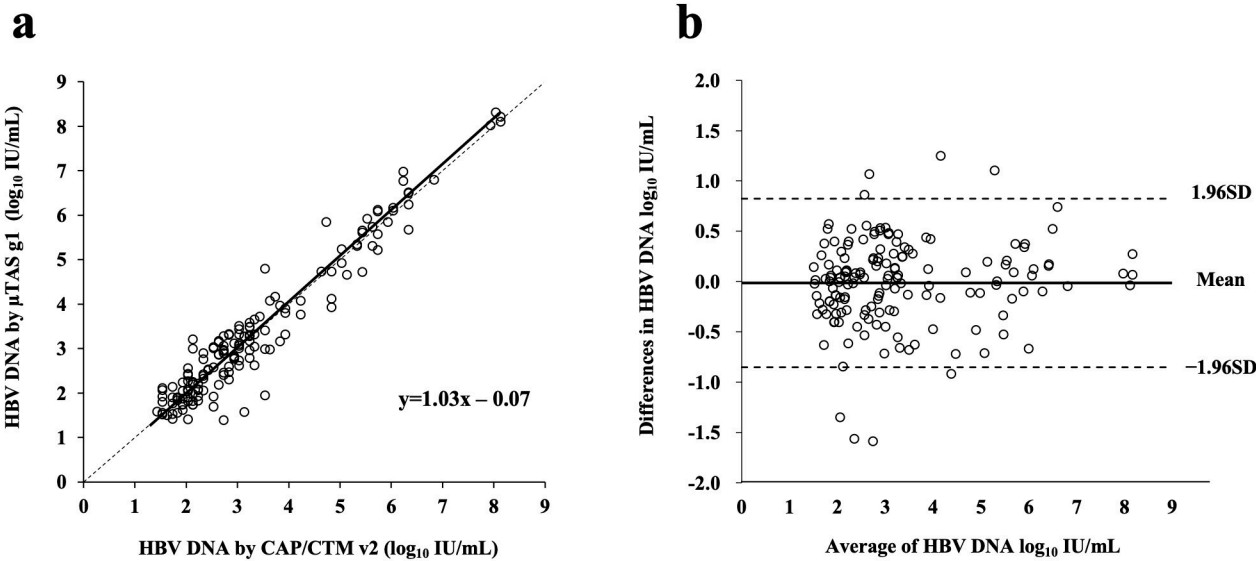

**Fig 4. Correlation of measurement and limit of agreement between HBV assays on the μTASWako g1 analyzer and CAP/CTM v2.** A total of 157 serum samples from HBV-infected patients were tested with CAP/CTM v2 and μTASWako g1 Analyzer. (a) Passing–Bablok regression fit (Solid line). Correlation coefficiency (*r*) was 0.964 (95% CI, 0.959–0.968), *P* < 0.0001. The dashed line represents the equality line. (b) Bland–Altman plot shows the mean difference between CAP/CTM v2 and μTASWako g1 Analyzer was −0.01 (limit of agreement, −0.82 to −0.85 $\log_{10}$ IU/mL). The solid and dashed lines represent the mean difference and mean ±1.96 SD, respectively.

## Discussion

We assessed the potential performance of the HBV DNA quantification assay using the μTASWako g1 analyzer. Results showed that the assay has a high sensitivity, specificity, and reproducibility across the quantification range from 20 to $10^8$ IU/mL. Precision of the μTASWako g1 HBV assay in the qualification range from 100 to $10^8$ IU/mL with SD of 0.02–0.11 $\log_{10}$ IU/mL and coefficient of variation of 0.33%–5.25%. In contrast, SD of the three existing commercial systems, CAP/CTM v2, Abbott RealTime HBV, and Aptima Quant HBV assay were 0.21 to 2.67, 0.44 to 11.25, 0.29 to 5.07, respectively, corresponding with the qualification range [19–22]. Thus, precision of the HBV assay using the μTASWako g1 analyzer was comparable to that of those systems. The LOD of the μTASWako g1 HBV assay (4.18 IU/mL and 4.35 for the serum and EDTA-plasma samples) were almost similar with that of the three existing systems based on the product information sheet (S3 Table).

The μTASWako g1 HBV assay utilizes IC to monitor the DNA purification process to PCR-CE detection. The amount of IC spiked into the samples was predetermined so that IC Cq can be reproducibly detected within a particular range when testing the sample with HBV DNA (up to $3.5 \times 10^8$ IU/mL) or without HBV DNA. Deviation of IC Cq from the predetermined range will indicate decreased DNA purification efficiency or the presence of substances that may inhibit the PCR amplification.

In clinical samples obtained from patients with chronic HBV infection, the HBV DNA levels measured using the μTASWako g1 were well correlated with those from CAP/CTM v2. Moreover, the average difference of reported HBV titers between μTASWako g1 and CAP/CTM v2 was relatively smaller when compared with the average difference in previous reports [21, 22]. In the Blant–Altman analysis, four (2.5%) and two (1.3%) samples had difference of >1.0 and 1.5 $\log_{10}$ IU/mL between the μTASWako g1 and CAP/CTM v2. One possible reason for the discrepancy might be the mutation in the PCR amplification region of the HBV genome. However, this study could not assess the sequencing for HBV primer target region

due to insufficient remaining sample volume for the analysis after the clinical study. The region targeted by primers of μTASWako g1 HBV assay are S gene, whereas the region targeted by primers/probes of CAP/CTM v2 are preC-C gene. A previous study reported deviation of HBV DNA levels measured by two different PCR assays, amplify different regions of HBV genome, occurring due to a sequence mismatched in the primer or probe in one of the assays [23]. HBV has been known to be a highly variable DNA genome and is divided into genotypes A to J based on the 8% sequence variations in the S gene [24]. We confirmed that the μTASWako g1 HBV assay could measure HBV DNA levels with accuracy of ±0.34 $\log_{10}$ IU/mL for HBV genotypes A to H. Additionally, *in silico* analysis was performed for primer homology with target region of the S gene for genotypes I and J [25]. The μTASWako g1 primer sequences were perfectly matched to that of the S gene in genotypes I and J (accession no. AB562463 [I1], FJ023669 [I2], AB486012 [J]). The HBV genome mutation was potentially induced by the long-term treatment of nucleoside analogs, such as lamivudine and adefovir [26–28]. Six clinical samples, with difference of >1.5 $\log_{10}$ IU/mL between μTASWako g1 and CAP/CTM v2, were obtained from patients without nucleoside analog treatment. Other mutations in the S gene induced by hepatitis B vaccine were well known [29, 30]. However, several reported mutation points were not associated with the primer regions of the HBV assay. Sequence analysis in the target region may be necessary to identify the reasons of different HBV DNA levels between the methods in clinical samples.

Overall, the assessment revealed that the μTASWako g1 HBV assay has a comparable performance to the existing HBV DNA assay systems. Furthermore, the assay can complete the measurement within 110 min. Therefore, the μTASWako g1 analyzer can provide measurement results to clinicians with minimal delay after blood collection and sample measurement compared to the existing commercial systems. This will be a strong advantage of this system, especially in monitoring patients at risk of HBV reactivation. However, two limitations were identified in the assay using the μTASWako g1 analyzer: (1) the assay throughput is four samples per run and (2) the sample is required for 1 mL of serum or EDTA-plasma sample per test. To enhance the usefulness of the assay for limited sample volume, further improvement of the assay performance may be required.

## Conclusions

The HBV assay on the μTASWako g1 analyzer is a sensitive and reproducible assay for HBV DNA quantification, which can provide test results within 2 h. The result also revealed a good correlation with the commercial HBV viral load assay, suggesting that the HBV assay on the μTASWako g1 analyzer potentially becomes an alternative method for HBV viral load test to help manage HBV infection in a timely manner.

## Supporting information

**S1 Table. Background information of clinical samples from patients with HBV infection for correlation study (n = 207).** Values shown are median (interquartile range).
(DOCX)

**S2 Table. The comparison of discrepancy between CAP/CTM v2 and μTASWako g1 assay in individual samples with lower HBV titer.**
(DOCX)

**S3 Table. Comparison of assay specification of CAP/CTM v2 assay, Abbott RealTime HBV assay and Aptima HBV Quant assay.**
(DOCX)

## Acknowledgments

The authors gratefully acknowledge Aurelie Souppe (FUJIFILM Wako Diagnostics USA) for supporting data acquisition and do Jian-Ping Zhang (FUJIFILM Wako Diagnostics USA) for advising on assay design.

## Author Contributions

**Conceptualization:** Tomohisa Kawabata.

**Data curation:** Moto Watanabe, Hidenori Toyoda.

**Investigation:** Moto Watanabe.

**Methodology:** Moto Watanabe.

**Supervision:** Hidenori Toyoda, Tomohisa Kawabata.

**Validation:** Moto Watanabe.

**Writing – original draft:** Moto Watanabe.

**Writing – review & editing:** Hidenori Toyoda, Tomohisa Kawabata.

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
