## [Decision Letter · Decision Letter 0]

13 Dec 2022

PONE-D-22-30604Rapid Quantification Assay of Hepatitis B Virus DNA in Human Serum and Plasma by Fully Automated Genetic Analyzer μTASWako g1PLOS ONE

Dear Dr. Kawabata,

Thank you for submitting your manuscript to PLOS ONE. After careful consideration, we feel that it has merit but does not fully meet PLOS ONE’s publication criteria as it currently stands. Therefore, we invite you to submit a revised version of the manuscript that addresses the points raised during the review process.

We look forward to receiving your revised manuscript.

Kind regards,

Md. Golzar Hossain, Ph.D.

Academic Editor

PLOS ONE

Journal Requirements:

"Specific grant numbers; None

Initials of authors who received each award; M.W

Full names of commercial companies that funded the study or authors; Fujifilm Wako Pure Chemical Corporation

Initials of authors who received salary or other funding from commercial companies; M.W and T.K

URLs to sponsors’ websites; https://www.fujifilm.com/ffwk/en

The funders had role in decision to publish of the manuscript."  

We note that one or more of the authors have an affiliation to the commercial funders of this research study: Fujifilm Wako Pure Chemical Corporation

(2) Please also provide an updated Competing Interests Statement declaring this commercial affiliation along with any other relevant declarations relating to employment, consultancy, patents, products in development, or marketed products, etc.  

Within your Competing Interests Statement, please confirm that this commercial affiliation does not alter your adherence to all PLOS ONE policies on sharing data and materials by including the following statement: ""This does not alter our adherence to  PLOS ONE policies on sharing data and materials.” (as detailed online in our guide for authors http://journals.plos.org/plosone/s/competing-interests). 

If this adherence statement is not accurate and  there are restrictions on sharing of data and/or materials, please state these. Please note that we cannot proceed with consideration of your article until this information has been declared.

Reviewers' comments:

Reviewer's Responses to Questions

**Comments to the Author**

1. Is the manuscript technically sound, and do the data support the conclusions?

Reviewer #1: Yes

2. Has the statistical analysis been performed appropriately and rigorously? 

Reviewer #1: Yes

3. Have the authors made all data underlying the findings in their manuscript fully available?

Reviewer #1: Yes

4. Is the manuscript presented in an intelligible fashion and written in standard English?

Reviewer #1: Yes

5. Review Comments to the Author

Reviewer #1: Comments to the authors

In this study, Watanabe et al. assessed the analytical and clinical performance of HBV DNA assay on the μTASWako g1 platform in human serum and EDTA-plasma. The authors found that the HBV DNA assay on the μTASWako g1 is potentially applied for alternative method of the HBV viral load test, in particular with the advantage of the HBV DNA result availability within 2h, improving the HBV infection management. The results of this study were interesting.

I have several comments.

1. In the methods section, authors described as follows: “Afterward, 33 PCR cycles with a longer extension (70 s at 73.5°C) was performed with one CE injection from the PCR chamber for each subsequent cycle to analyze the accumulated target amplicons and took 56 min to complete the entire qPCR-CE step.” How did you sample from the PCR chamber into the electrophoresis channel?

2. Normally in PCR reactions, it is thought that the amplification reaction efficiency changes with changes in the concentration of the multiplex partner when the reaction is multiplexed, but what is the reason why the IC reactivity did not change when the HBV virus concentration was changed?

3. How can this "quick results" measurement method be useful in real-world clinical practice? Please discuss this point.

6. PLOS authors have the option to publish the peer review history of their article (what does this mean?). If published, this will include your full peer review and any attached files.

Reviewer #1: No

---

## [Author Response · Author response to Decision Letter 0]

10 Jan 2023

To Reviewer

Question 1:

 In the methods section, authors described as follows: “Afterward, 33 PCR cycles with a longer extension (70 s at 73.5°C) was performed with one CE injection from the PCR chamber for each subsequent cycle to analyze the accumulated target amplicons and took 56 min to complete the entire qPCR-CE step.” How did you sample from the PCR chamber into the electrophoresis channel?

Answer: 

PCR product (DNA) is moved from PCR chamber toward electrophoresis channel by applying electrical current (0.2 μA) to PCR chamber via two microchannels which filled with a reagent for electrophoresis. The reagent wells were physically separated from electrode contact wells to prevent instrument contamination from reagent and amplicons. Electrical connection between the two wells was made through printed carbon electrodes on the chip. The detail of microfluidic chip design and the movement of DNA during the CE were well illustrated in previous research article (reference#18). The article is also helpful to understand how it works.

Question 2:

 Normally in PCR reactions, it is thought that the amplification reaction efficiency changes with changes in the concentration of the multiplex partner when the reaction is multiplexed, but what is the reason why the IC reactivity did not change when the HBV virus concentration was changed?

Answer: 

There are three reasons why the IC reactivity did not change when the HBV virus concentration was changed.

1. In the μTASWAKO g1 HBV assay, IC DNA (approximately 1,200 copies/assay) was added to the sample at the beginning of the DNA purification step, therefore, the IC DNA concentration in PCR mixture is almost consistent every times.

2. IC DNA was designed so that IC target sequence can be amplified by HBV primers. Basically, the PCR amplification efficiency of IC were equivalent to that of HBV target due to the same primer performance. 

3. The IC DNA concentration, the primer concentration, and IC detection threshold for qPCR-CE have been optimized to reproducibly detect IC Cq by testing the samples containing high concentration of HBV (~3.5×10^8 IU/mL). The balance of those three factors is most important to enable IC detection without interference by HBV-amplification even when 3.5×10^8 IU/mL of HBV sample tested. Primer consumption in a reaction-mixture results in reduction of amplification efficiency as a progressing of PCR cycles. In the case of co-amplification of HBV (3.5×10^8 IU/mL) and IC, the HBV primers were consumed, however, the remaining primers maintains IC-amplification efficiency until PCR cycle (at 26-27) when IC-amplicons exceed the detection threshold for qPCR-CE. We have confirmed that IC Cq does not change across dynamic range of HBV quantification (~3.5×10^8 IU/mL), while IC Cq may change when testing of the sample containing HBV more than 10^9 IU/mL. If the assay detected HBV DNA at over dynamic range, the sample will need to be diluted with HBV-negative sample for re-test. 

Question 3: 

How can this "quick results" measurement method be useful in real-world clinical practice? Please discuss this point. 

Answer: 

HBV is potentially reactivated in some patients who undergo chemotherapy or immunosuppressive therapies and HBV reactivation often causes life-threatening liver failure. Reactivation occurs not only in HBsAg-seropositive patients but also in those with resolved HBV infection who are seronegative for HBsAg but seropositive for antibody against hepatitis B core antigen (anti-HBc) and/or antibody against HBsAg (anti-HBs). To prevent HBV reactivation-related hepatitis, HBV DNA monitoring-guided preemptive antiviral therapy using anti-HBV nucleos(t)ide analogs (NAs) is recommended by several guidelines for patients with resolved HBV infection, and it is mandatory to detect the emergence of serum HBV DNA as early as possible to start anti-HBV therapy early and effectively. However, the duration required for HBV DNA measurements, including both qualitative and quantitative, hinders agile and effective preventive strategies in real clinical practice. HBV DNA measurement technology with rapidly available results will be a strong advantage for monitoring patients at risk of HBV reactivation since the “quick results” possibly contribute to minimal delay in clinical decision-making after blood collection from the patients.

Best regards,

Tomohisa Kawabata

---

## [Decision Letter · Decision Letter 1]

23 Jan 2023

Rapid Quantification Assay of Hepatitis B Virus DNA in Human Serum and Plasma by Fully Automated Genetic Analyzer μTASWako g1

PONE-D-22-30604R1

Dear Dr. Kawabata,

We’re pleased to inform you that your manuscript has been judged scientifically suitable for publication and will be formally accepted for publication once it meets all outstanding technical requirements.

Kind regards,

Md. Golzar Hossain, Ph.D.

Academic Editor

PLOS ONE

Additional Editor Comments (optional):

Reviewers' comments:

Reviewer's Responses to Questions

**Comments to the Author**

1. If the authors have adequately addressed your comments raised in a previous round of review and you feel that this manuscript is now acceptable for publication, you may indicate that here to bypass the “Comments to the Author” section, enter your conflict of interest statement in the “Confidential to Editor” section, and submit your "Accept" recommendation.

Reviewer #1: All comments have been addressed

2. Is the manuscript technically sound, and do the data support the conclusions?

Reviewer #1: Yes

3. Has the statistical analysis been performed appropriately and rigorously? 

Reviewer #1: Yes

4. Have the authors made all data underlying the findings in their manuscript fully available?

Reviewer #1: Yes

5. Is the manuscript presented in an intelligible fashion and written in standard English?

Reviewer #1: Yes

6. Review Comments to the Author

Reviewer #1: Comments to the Authors

I have carefully reviewed your replies. You have responded to all questions appropriately. I did not think any further revision was necessary.

7. PLOS authors have the option to publish the peer review history of their article (what does this mean?). If published, this will include your full peer review and any attached files.

Reviewer #1: No

---

## [Editor Report · Acceptance letter]

31 Jan 2023

PONE-D-22-30604R1 

Rapid Quantification Assay of Hepatitis B Virus DNA in Human Serum and Plasma by Fully Automated Genetic Analyzer μTASWako g1 

Dear Dr. Kawabata:

I'm pleased to inform you that your manuscript has been deemed suitable for publication in PLOS ONE. Congratulations! Your manuscript is now with our production department. 

Kind regards, 

on behalf of

Dr. Md. Golzar Hossain 

Academic Editor

PLOS ONE